chemical physics/energy/mechanics

elasticity, damping ratio, super-hydrophobic surfaces, robustness, viscosity

**Author for correspondence:**
Meirong Zhao
e-mail: meirongzhao_acad@163.com

# Elasticity and damping ratio measurement of droplets on super-hydrophobic surfaces

Yukai Sun[1], Yelong Zheng[1], Le Song[1], Peiyuan Sun[2], Meirong Zhao[1], Yixiong Zhou[3] and Clarence Augustine TH Tee[4]

[1]State Key Laboratory of Precision Measuring Technology and Instruments, Tianjin University, Tianjin, People's Republic of China
[2]Dongying Vocational Institute, Dongying, People's Republic of China
[3]Department of Ophthalmology, Ninth People's Hospital, Shanghai Jiao Tong University School of Medicine, Shanghai, People's Republic of China
[4]Department of Electrical Engineering, Faculty of Engineering, University of Malaya, Kuala Lumpur, Malaysia

MZ, 0000-0003-3027-8065

The measurement of the droplets' elasticity is vitally important in microfluidic and ink-jet printing. It refers to the ability of the droplet to restore its original shape and strong robustness. This study investigated a novel method to measure elasticity. The plate coated with super-hydrophobic layers pressed on a droplet and the elastic force was recorded by an electronic balance. Meanwhile, a mathematical model was constructed to calculate the changes of the droplet area under the force. The measurement showed that external work mainly converts into surface energy and the damping ratio increases from 0.068 to 0.261 with the increase of mass fraction from 0 to 80 wt%. It also indicates that the novel method can accurately and efficiently measure the elasticity of droplets.

## 1. Introduction

The elasticity of droplets is important in many natural processes [1] and industrial applications [2], such as anti-icing [3], ink-jet printing [4] and targeted delivery [5], which has a broad range of applications from biomedical technologies [6,7] to personal care. For example, the targeted delivery can transport a drug to the desired location, and release it on demand [8,9]. Currently, micro-fluidic systems can control the formation of droplets using electric and magnetic fields automatically. And droplet elasticity represents the effectiveness in controlling the droplets [10]. It also indicates the capability to resist high pressure, impact and deformation, which is particularly crucial

for the transportation and manipulation of the droplet in chemical reactions, biomedical sensors and optical detection [11].

Recently, a series of experiments [12–14] have been designed to directly measure the elasticity of the droplet. For example, when a drop of liquid is coated with hydrophobic powders [15], which could turn into a completely non-wetting droplet known as a liquid marble, place it directly between the electronic balance and the plate. The liquid marble elasticity can be analysed using the relationship between its force value and the pressing distance. Due to the asymmetry of the surface particles, there exists a difference in the elasticity of the identical volume marble [11,16,17]. Another measurement method is to measure the elasticity of the liquid marble by dropping the droplet from a specified height and observing its final state of motion [18,19]. When the height is low (at small impact velocity), the droplet bounces on the substrate. As the height increases (at higher impact velocity), the droplet ruptures. In other words, this method can be used to measure the elasticity of droplets within a certain range. It means that when the Weber number ($We = (\rho V^2 D)/\sigma$, where $\rho$, $D$, $V$, $\sigma$ are the droplet density, diameter, impact speed, and surface tension, respectively) exceeds a certain value, the current method cannot be used to measure the elasticity of the droplet. When the droplet impacts the surface, its kinetic energy flattens the droplet and converts it into surface energy. As the impact velocity increases, more kinetic energy is consumed in droplet deformation and less is used for bouncing. Thus, a complete rebound is at higher possibility with lower Weber numbers. While at larger Weber numbers, partial rebound or even satellite phenomena happen where droplets break up [20–22]. Moreover, this method is also limited by the volume of droplets.

To measure the elasticity of the droplet and analyse the factors affecting the elasticity of the droplet without restrictions, we drew lessons from the measurement method of liquid marble. There are currently two methods to achieve non-stick between the droplet and solid: the first is a droplet densely covered with hydrophobic particles and the second is a super-hydrophobic substrate that ensures a large contact angle between the droplet and the substrate. The super-hydrophobic substrate is a new multi-functional surface and has driven possible applications such as self-cleaning and ice resisting [23]. In nature, such surfaces could be found on, among others, lotus leaves [24], butterfly wings and cicada's wings [25]. It exhibits nanoscale outer-layer wax with low substrate energy. Recently, more techniques have been found to add super-hydrophobicity to a surface, including spray coating [26] and the layer-by-layer process [27]. Those two methods are commercially available for practical applications due to their better chemical and mechanical stabilities [28].

Here, droplets of different volumes and viscosities were placed between two parallel plates of super-hydrophobic substrates. The lower plate was tightly attached to the electronic balance, and the upper plate pressed down at a controlled speed. With the decreased distance, the indication of the electronic balance was captured by the console. It was found that the force to deform water droplets had a similar trend with the change in droplet surface area, no matter how the volume changed. Moreover, it can be inferred that the elasticity of the droplet was provided by the surface tension, which could also prove the correctness of our experimental method. Overall, our experiment drew on the advantages of liquid marble elasticity measurement and introduced a functional interface of the super-hydrophobic surface. This may be a new solution for the measurement of droplet elasticity and broke the limitation of the Weber number and volume, providing technical support for the application of droplets in microfluidics, sensors and so forth.

## 2. Material and methods

The super-hydrophobic substrate (figure 1a) was made using the NeverWet (Ross Technology Corp) and the macroscale roughness of the surface is obvious (scale bar 1 μm). A droplet (figure 1b) of 3 μl mixed with purple pigment on a super-hydrophobic substrate and the contact angle is $160 \pm 4°$. The droplet deposited on the surface can reach a position of equilibrium between gravity and capillary forces. Before the experiment, both the pressure plates were uniformly coated with hydrophobic particles to ensure the hydrophobicity of the substrate. This can eliminate the interference of the substrate, prevent liquid loss, maintain the droplet spherically and restore the original shape of the droplet.

For verifying the relationship between the elasticity of droplets and viscosity, we prepared three different concentrations of glycerol solution (AR, Shanghai Hushi Laboratorial Equipment Co., Ltd.): 0 wt%, 50 wt% and 75 wt%. The viscosity of glycerol solution with different concentrations varies greatly, but the difference of surface tension can be neglected [29]. For example, the surface tension values of 0 wt%, 50 wt% and 75 wt% glycerol solutions are 72.8 mN m$^{-1}$, 68.3 mN m$^{-1}$ and

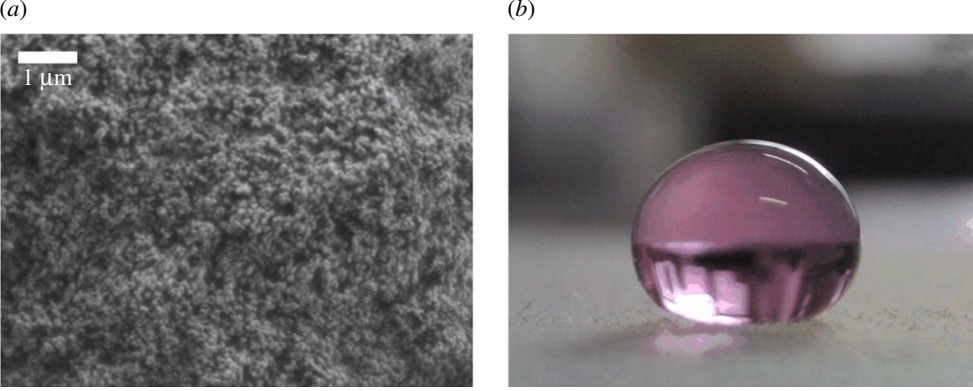

**Figure 1.** Materials. (*a*) Scanning electron micrograph of super-hydrophobic substrate. (*b*) Side view of 3 μl droplet mixed with purple pigment on the super-hydrophobic substrate.

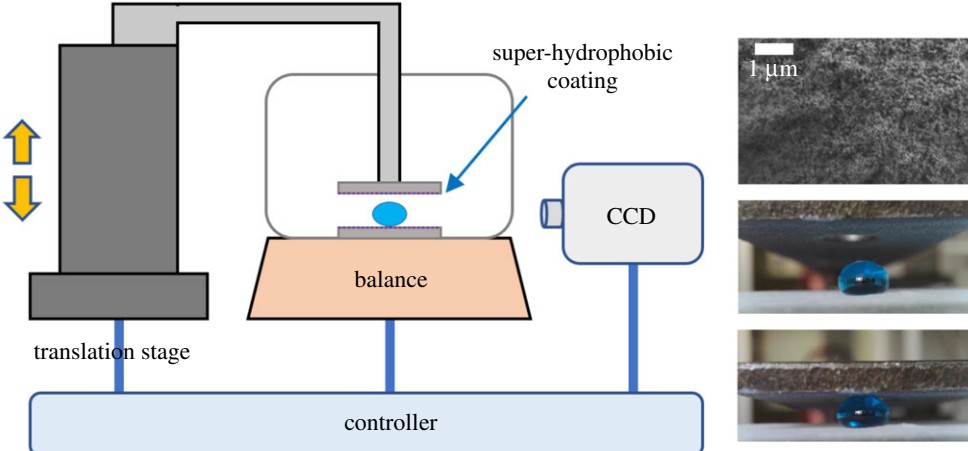

**Figure 2.** Diagram of devices for measuring elasticity of the droplet on super-hydrophobic surface. A droplet is squeezed by two parallel plates at a constant speed. The balance and CCD are used to monitor the compression indication and morphology of the droplet, respectively.

65.3 mN m$^{-1}$, respectively, but the viscosity has a big disparity of 1.0 mPa s, 6.0 mPa s and 60.0 mPa s. We can assess how the viscosity affects the droplet elasticity by measuring the elasticity of the same volume but with different concentrations of glycerol droplets under the same compression distance. Glycerol solutions were prepared with a stirrer to fully mix the water and glycerol stock solution. Different types of glycerol solution in the fixed volume were deposited with a precision pipette onto the super-hydrophobic surface (figure 2). Compared with elasticity values, we can analyse the influence of volume, viscosity and surface tension on the elasticity. One group was set as the experimental group, and the other two groups were set as the control groups to verify those effects.

In the first place, the top and side views of the different volume droplets were taken separately. The height and horizontal diameters were then measured and analysed using ImageJ software. The individual drop was compressed between two plates in a custom apparatus that used a balance (JA3003J-SOPTOP) and an electric displacement platform (OSMS60-10ZF-SIGMAKOKI) to measure the force *F* required to compress the drop at a distance. The upper and lower plates were coated with the super-hydrophobic coating to prevent water infiltration and maintain the contact angle at 160°. Importantly, the experimental facility allowed for observation and documentation from the side during compression using a camera (ANDONSTAR-A1). All the experiments were carried out at room temperature (25°C) and relative humidity of 54%.

Three types of droplets were placed on super-hydrophobic surfaces using a precision pipette at volumes from 5 to 90 μl at 5 μl intervals. During compression, the top plate was lowered at a rate of 10 μm s$^{-1}$ while both the drop deformation and the reading on the balance were digitally recorded. Balance readings were immediately converted to force values and synchronized with the picture from

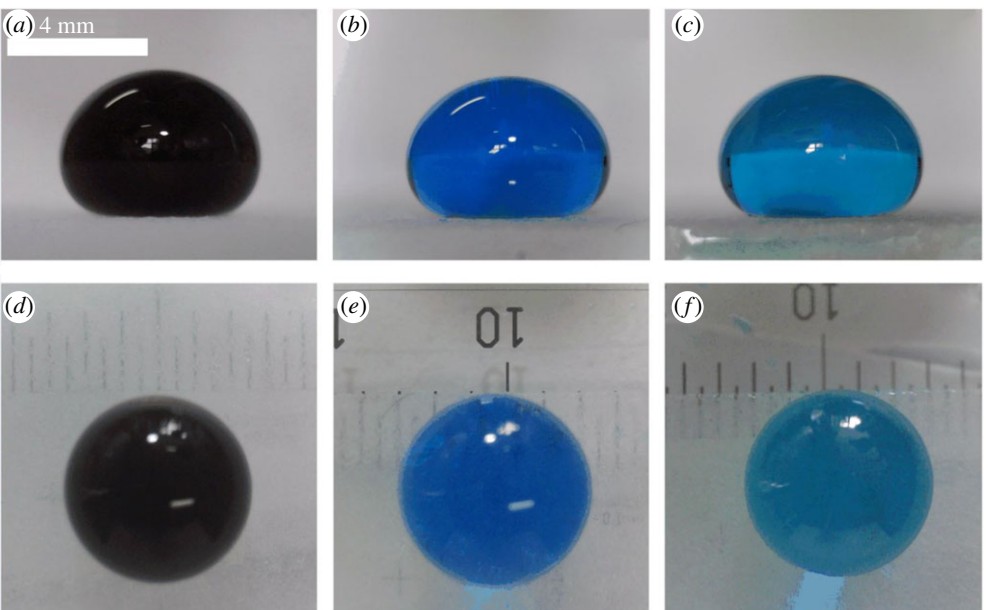

**Figure 3.** Side views of 45 µl glycerol droplets with different concentrations: (*a*) 0 wt%, (*b*) 50 wt%, (*c*) 75 wt%. Top views of 45 µl glycerol droplets with different concentrations: (*d*) 0 wt%, (*e*) 50 wt%, (*f*) 75 wt%. The mixed solutions of the three concentrations were used in the elasticity measure.

the CCD. The droplet size was measured using ImageJ (National Institutes of Health, Inc) and data analysis was performed in MATLAB (The MathWorks, Inc.).

# 3. Results

## 3.1. Static shape of droplets on the super-hydrophobic surfaces

To study the height variation of a droplet with the water volume, we used the parameter $H$ to define its height. As the volume increased, the shape of the droplet changed from quasi-spherical to a puddle shape (figure 3). It was found that the height difference could be ignored despite the equal volume droplet formed by liquids of different concentrations. Moreover, the height of droplets formed by the identical volume was the same. Figure 4 shows that the process can be divided into two regions. In region 1, the shape of the droplet is only determined by the surface tension ($H < k^{-1} = \sqrt{\sigma/\rho g}$, where $\sigma$, $\rho$ and $g$ are the droplet surface tension, density and acceleration of gravity, respectively). Due to the droplet weight, the droplet tends to lower its centre of mass and forces a contact. The surface tension opposes the formation of this contact and increases the droplet area [30]. The surface tension difference of the three types of droplets is relatively small and the Laplace pressure ($\Delta P = 2\sigma/r$, where $r$ is the droplet radius) causes the droplet to keep in spherical condition, which is closer to the one derived from the spherical formula. In zone 2 ($H > k^{-1}$), gravity and surface tension work together on the droplets. It remains spherical but has little deformation. As the volume increases, gravity is the dominant factor affecting the deformation. The shape begins to deform due to gravity and the height begins to change slowly with volume, but the height difference of different types of droplets can be ignored, which provides conditions for elasticity measurement. For a large volume height of a liquid puddle, $H$ can be expressed as [31]:

$$H = 2k^{-1} \sin\left(\frac{\theta}{2}\right). \tag{3.1}$$

Here, $k^{-1}$ is the capillary length. If supposedly a droplet contact angle $\theta$ is 180°, the maximal height of $H$ tends asymptotically to be twice the capillary length [32]. Thus, the maximum height, $H_{max}$ can be calculated as

$$H_{max} = 2\sqrt{\frac{\sigma}{\rho g}}. \tag{3.2}$$

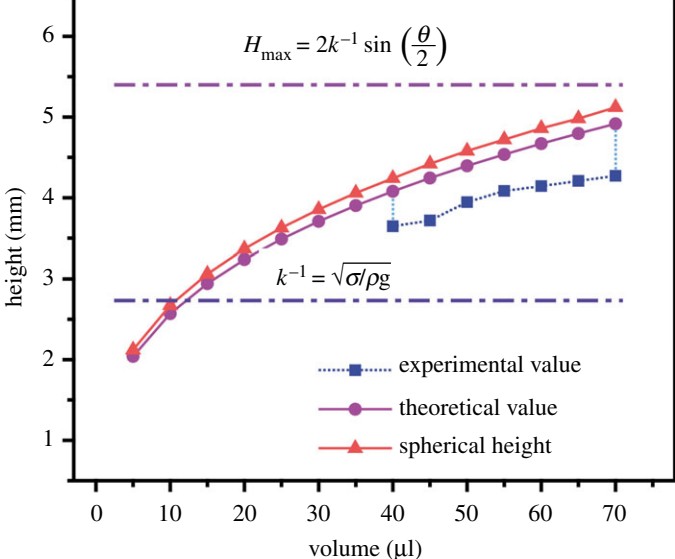

**Figure 4.** Droplets height versus volume. Under $k^{-1}$, the height of the droplet is only determined by the surface tension ($B_0 = (\rho g\, R^2)/\sigma < 1$). Exceed $k^{-1}$, gravity and surface tension work together on the droplets. It remains spherical but has a small deformation. As the volume increases, gravity is becoming the dominant factor affecting deformation. The shape begins to deform due to gravity and the height begins to change slowly with volume. Due to the drop weight, the drop tends to lower its centre of mass and forces a contact. But surface tension opposes the formation of this contact and increases the droplet area.

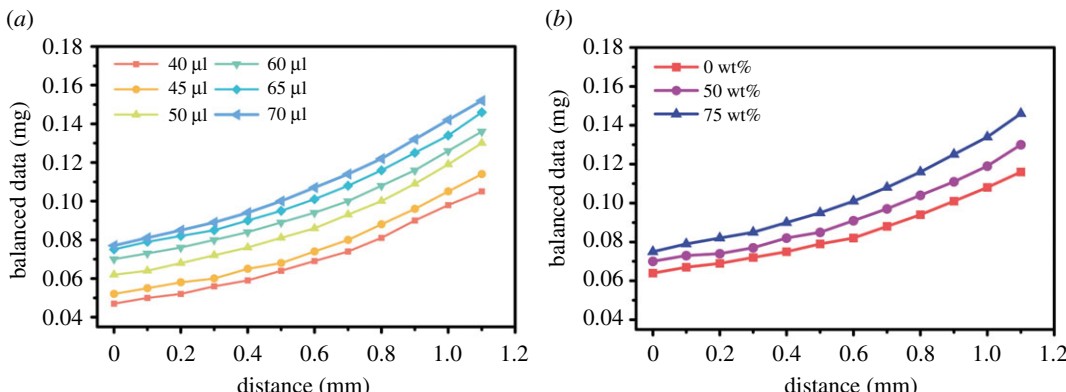

**Figure 5.** The balance reading curve diagram in the compression process. (*a*) The mass values recorded by the electronic balance when 50 wt% glycerol droplets with different volumes are squeezed at different distances. The starting point of the curve is the gravity of the droplet. With the increase of the droplet volume, the mass values also increase and the curve moves up in the vertical direction. (*b*) The elasticity at different viscosity values. Droplets with volumes of 45 μl were prepared using glycerol with concentrations of 0 wt%, 50 wt% and 75 wt%.

Equation (3.2) could explain why the height does not change as the volume increases, since the height of the droplet is determined by surface tension, density and gravitational acceleration but has no relationship with volume.

## 3.2. Elastic measurement of droplets on the super-hydrophobic surfaces

The experimental process (figure 3) was carried out and the electronic balance reading curve as shown in figure 5. The data collection process of the electronic balance was as follows:

(a) The initial position (before the plate deformed the droplet): the plate moved down towards the droplet at 10 μm-distance step.
(b) After the droplet was stationary for 5 s, the electronic balance collected the data and sent it to the computer. To ensure the accuracy and synchronization of the experiment, Labview was used to control the operation of the equipment and eliminate any errors caused by manual control. The

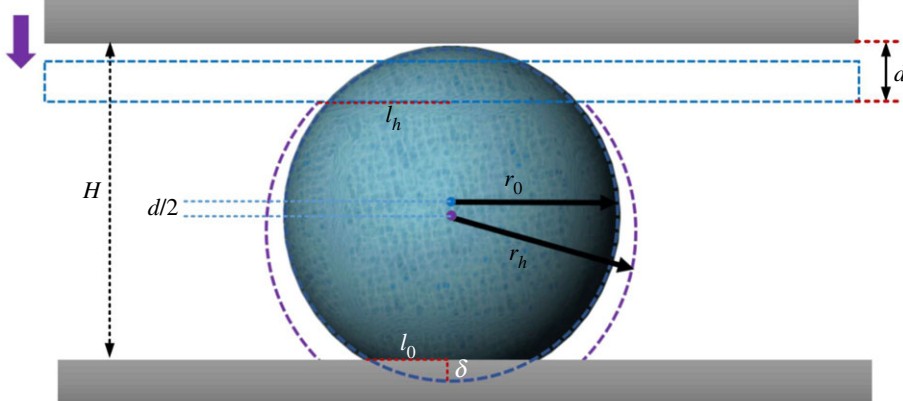

**Figure 6.** Compression model schematic diagram of a droplet.

purpose of collecting data after the droplet was stationary for 5 s was to eliminate the interference from droplet bounce.

(c) When the plate was pressed down, part of the energy would be converted into the kinetic energy of the droplets. After 5 s, the kinetic energy would be completely converted into the surface energy of the droplet.

We can infer from figure 5a that the data recorded by the electronic balance have the same trend when the droplet with the same solution was pressed at the same displacement, with volume from 40 to 70 μl. In other words, the coefficient of elasticity is independent of the volume. The starting point of the curve is the mass of the droplet. The elasticity of the droplet can be obtained by reducing the weight of the droplet. The indicator on the electronic balance is the sum of the forces of gravity and elasticity. When the plate is not in contact with the droplet, the droplet gravity can be obtained by the electronic balance.

Because the droplet volume does not effect the elasticity, there must exist a relationship between the elastic coefficient and the deformation: the elastic coefficient increases gradually with the increase of deformation. Elastic force is provided by the internal pressure, which is balanced with the pressure produced by the plate. In the process of extrusion, the radius of the droplet and the contact area change, which affects the elasticity of the material. The contact area of small droplets is different from large droplets when they are compressed at a certain distance. For example, the 40 μl droplet contact area is smaller than the 70 μl droplet.

To assess whether the viscosity of the liquid affected the elasticity of the droplet, the following experiment (figures 5b) was carried out in which the same volume of the droplet with different concentrations was pressed and the elastic growth rate of the droplets was different.

# 4. Discussion

## 4.1. Surface tension and volume

The 'squeezed' droplet model (figure 6) was established to explain the surface tension and volume affecting the elasticity of the droplet. It could predict the change of droplet surface area with different compression.

The $H$ represents the height of the droplet in a static state, $r_0$ represents the radius when the droplet is spherical and $l_0$ represents the radius of the contact surface. When the contact angle of the hydrophobic material is 180°, the radius of the contact area between the droplet and the interface is close to 0, which is a completely spherical shape. In the actual experiment, the contact angle of the hydrophobic material is $160 \pm 4°$. Under the effect of gravity, the contact radius is usually greater than 0, and the height of the centre of gravity drops by $\delta$. The equation, $\delta = \sqrt[3]{3V/4\pi} - H$, indicates the drop height due to gravity ($d$ represents the height of the pressure plate descending, $r_h$ represents the radius of the hemispherical after being squeezed and $l_h$ represents the radius of the upper contact surface after squeezing). The top of the contact surface is a perfect spherical shape, and the contact surface is a circle with a radius of $l_0$: $l_0 = \sqrt{r_0^2 - (r_0 - \delta)^2}$. When the upper-pressure plate is in contact with the droplet, the contact

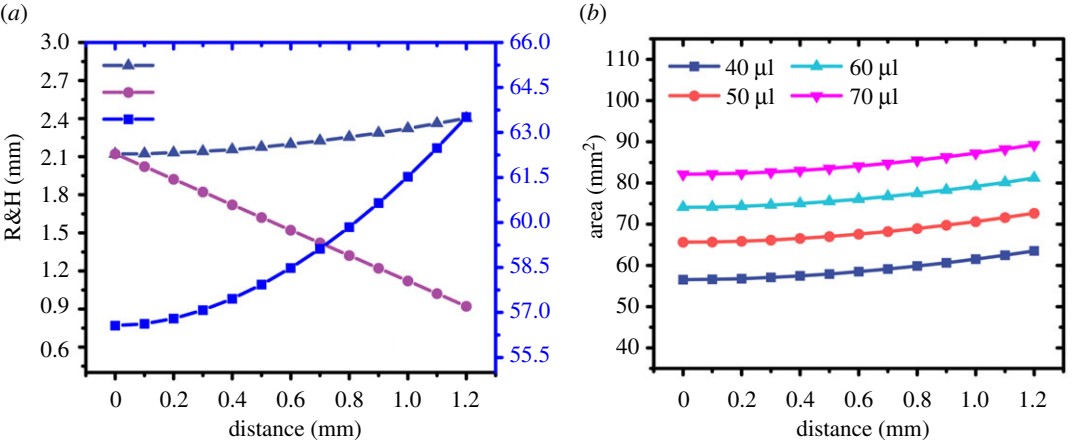

**Figure 7.** (a) The radius, height, area curve of the 70 μl droplet during compression. (b) The area curve of the different volume droplet during compression. As the distance between the two pressure plates decreases, the radius of the droplet increases slowly, but its surface area increases rapidly. The trend is similar to the trend of elastic growth. In the same compression distance for droplets of different volumes, the difference can be ignored.

surface of the upper and lower part of the droplet with the pressure plate is approximately equal, which is $l_0 = l_h$. During the extrusion process, the droplet would deform and the upper and lower contact surfaces have the same area. When the pressing plate presses down $d$, the area of the upper and lower contact surfaces ($S_{top}$, $S_{bottom}$) can be represented by equation (4.1):

$$S_{top} = S_{bottom} = \pi r_h (d + \delta) - \pi \left[ \frac{(d + \delta)}{2} \right]^2. \tag{4.1}$$

When the $d$ is constant, $S_{top}$ is proportional to $r_h$: $S_{top} \sim r_h$. And the pressure is inversely proportional to $r_h$: $\Delta P_h \sim 1/r_h$. It can be seen that the pressure is eliminated by the droplet radius, and the pressure does not correlate with the volume. The pressure is mainly related to the extrusion distance $d$. During the squeezing process, the volume ($V$) of the droplet remains unchanged and the pressure applied on the upper and lower parts is the same. $V_{top}$ is the upper part, $V_{bottom}$ is the lower part and $V$ is the total volume of the droplet. $V_h$ represents the complete spherical dimensions of the droplet with the relationship equation as:

$$V_h - V_{top} - V_{bottom} = V_h - 2V_{top} = V \tag{4.2}$$

and

$$\frac{4}{3}\pi r_h^3 - 2\pi \left( \frac{d + \delta}{2} \right)^2 r_h + \frac{2}{3}\pi \left( \frac{d + \delta}{2} \right)^3 = V. \tag{4.3}$$

The droplets volumes used in the experiment were 5 μl to 90 μl. The plate moved at a fixed speed of 10 μm s$^{-1}$, and the movement distance was 1 mm. During the compression process, the area of the droplet changed. Combining the radius obtained in the volume formula above, the area formula of the spherical gap can be written as

$$S_h = S_2 - 2S_1 + 2S_{top} \tag{4.4}$$

and

$$S_h = 4\pi r_h^2 - 4\pi r_h \left( \frac{d + \delta}{2} \right) + 2\pi r_h (d + \delta) - \pi \frac{(d + \delta)^2}{2}. \tag{4.5}$$

The radius and area of the compressed droplet can be obtained through simulation. Figure 7a shows the simulation result about the area and radius of the droplets with the increasing of the pressing distance. Figure 7b shows the area curve of the different volume droplets during compression. Through a series of experiments in compressing glycerol droplets, we found that the elasticity of the droplet is mainly related to the surface tension of the droplet, but not much related to the volume.

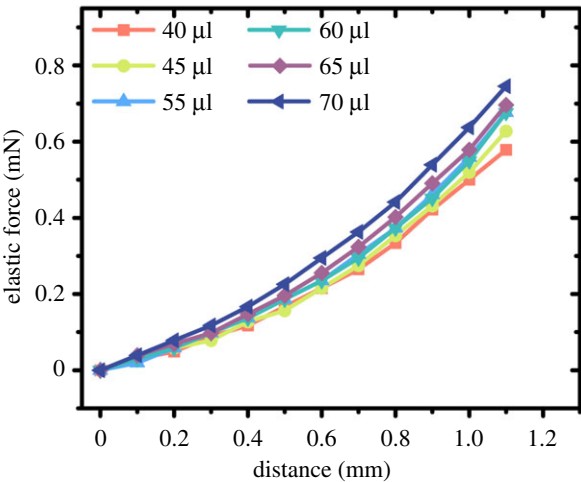

**Figure 8.** 75 wt% glycerin solution elasticity curve of different volume droplet. The little difference in curve is due to the volume diversity, which could be ignored.

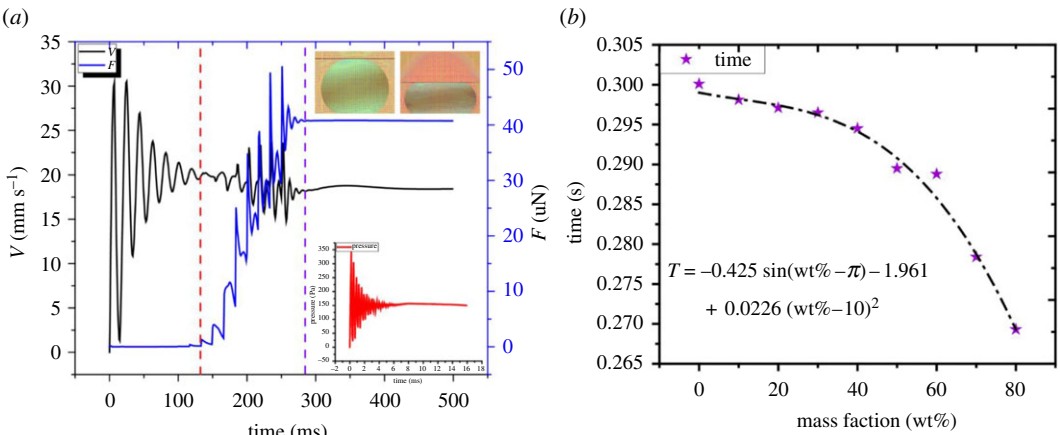

**Figure 9.** (*a*) During the simulated experiment, the pressure on the plate, the internal static pressure and velocity of the droplet are shown. (*b*) The time required for a droplet of different viscosity to be compressed for the same distance in stabilizing the droplet.

The growth rate of the elastic coefficient was a quadratic function curve, which was related to the area change rate of the sphere during the extrusion process. The indication of the balance was the combined action of the gravity of the droplet and the elastic force generated by the pressing plate. With the removal of the mass of the droplet itself, the elasticity of the droplet was retained. The data is shown in figure 8.

The balance was subjected to the pressing force of the plate and gravity in the vertical direction. For the droplet, when subjected to the pressure in the vertical direction of the parallel plate, the pressure in the opposite direction would be generated inside the droplet. The size was the same and the droplet was stably located between the two plates in a balanced state. Since the absolute parallelism of the two plates cannot be guaranteed during the experiment, the pressure plate would produce a partial force in the horizontal direction. When the force was sufficient to overcome the friction between the plates and droplet, it would leave the original position and the indication decreases sharply, which marks the end of the extrusion process.

During the experiment, the plate moved every 5 s and recorded the indications of the electronic balance. The 5 s was used to eliminate the interference of the kinetic energy inside the droplet. In this period, the kinetic energy can fully convert into the surface energy of the droplet.

## 4.2. Viscosity

A simulated experiment was set up to make a plate with a hydrophobic angle of 160° to squeeze a 5 μl drop at the speed of 3 mm s$^{-1}$. In figure 9*a*, we can deduce that: the pressure plate began to contact the

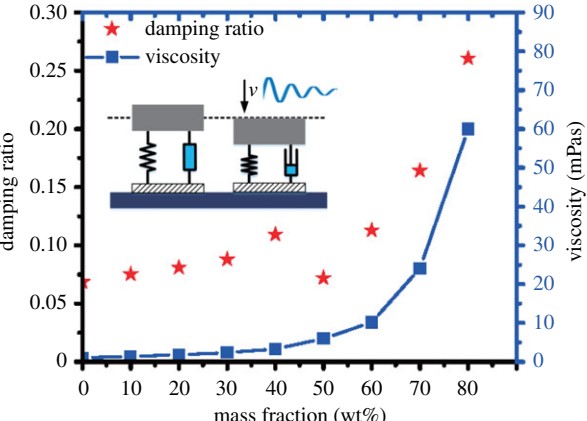

**Figure 10.** The standard mass-spring-damper (MSD) model. The relationship between damping ratio and droplet viscosity.

droplets at 140 ms, and the plate stopped moving at 250 ms. The total movement of the pressure plate was 600 μm. After 280 ms, the internal pressure remained at 180 Pa. The contact force and the velocity of the plate were also stable. Moreover, the energy by the plate was converted into the surface energy and the gravitational potential energy of the droplet. The 5 s time frame can guarantee the energy conversion process was completed during the experiment.

The droplets with different viscosity [33,34] would consume different times to make the pressure inside the droplet stable. The simulated data (figure 9b) show that the time for internal pressure stability is different when the mass fraction of glycerol solution changes. When the mass fraction of the liquid is 10 wt%, the viscosity is 1.4 mPa s, but when the mass fraction increases to 80 wt%, the viscosity is 60 mPa s. The increasing trend of viscosity with the mass fraction is inversely proportional to the internal stability time of the liquid. Consequently, viscosity is the main factor determining internal stability.

In order to evaluate the influence of droplet viscosity on damping, we have analysed the data collected [35]. When the platen is contacted with the droplet and stopped extrusion, its trampolining response can be considered a standard mass-spring-damper (MSD) system. It can be described by the following formula as [36]:

$$\frac{\mathrm{d}^2 y}{\mathrm{d}t^2} + 2\varsigma \frac{\mathrm{d}y}{\mathrm{d}t} + f_k(t) = -B_0, \qquad (4.6)$$

where $B_0 = mg/\sigma R_0$ ($R_0$ is the initial droplet radius, $\sigma$ is the surface tension and $B_0$ is the Bond number) and $\varsigma$ is the damping ratio. We distinguish three cases: $0 < \varsigma < 1$ (under-damper), $\varsigma = 1$ (critically damped) and $\varsigma > 1$ (over-damped) [37]. In our MSD system, $\varsigma < 1$ and the droplet in an under-damper condition.

The droplet can be considered as a standard MSD system, and the elastic potential energy is mainly provided by the hydrophobic substrate compressed droplet. The energy by the pressure plate is converted into the elastic potential energy of the droplet surface. And the damping force is mainly generated by the viscous force of the liquid. The damping ratio of different droplets can be calculated by measuring the amplitude attenuation of droplets motion in an unstable state. When the mass fraction of liquid increased from 0 to 80 wt% (figure 10), the damping ratio increased from 0.068 to 0.261, and the viscosity of the liquid increased from 1 to 60 mPa s.

The average values of multiple lines (figures 7b and 8) could be used to express the overall trend since the area difference between the volumes can be ignored (when $d = 600$ μm, $\Delta S < 0.049$ mm$^2$). We used the Taylor formula to fit the average values. The fitting curve (figure 11a) performed on the shape data (RMSE = 0.02022). And the elastic force (figure 11b) also can be expressed by a quadratic function with the increase of compression distance (RMSE = 0.005486). The result mainly shows that the elasticity of the droplet is mainly provided by the surface tension, and the droplet volume and viscosity has little effect on the elasticity. Future work will focus on the design of experimental setups in exploring the usage of droplets elasticity for innovative application and production.

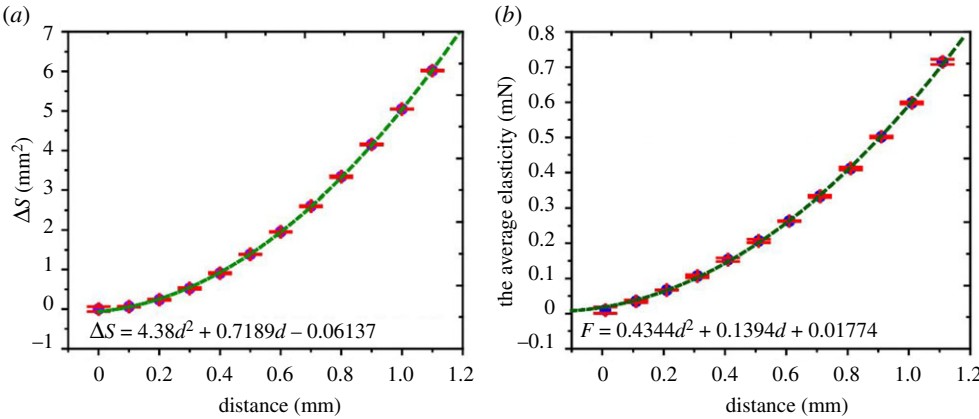

**Figure 11.** The experimental simulation work of the change of surface area and elastic force of droplets versus the compression distance.

## 5. Conclusion

This paper provided innovative methods to measure the elasticity and the results verify the viscosity and volume of the droplet have little effect on the elasticity but the surface tension is the main determinant to the elasticity. Meanwhile, the static properties and damping ratio range of droplets provide conditions for measurement. In the first place, the height of the different volume droplets with different mass fractions on the hydrophobic layer was obtained by the image method. It was found that the height of droplets of the different mass fractions was the same for the identical volume. Also, an electronic balance recorded measurement data by varying the distance of the upper and lower slices coated with the hydrophobic coating. The simulation experiment of droplet compression proved that it was obligatory to wait for 5 s to collect data during the experiment. Moreover, the damping ratio increased from 0.068 to 0.261 with the mass fraction of liquid increased from 0 to 80 wt%. In addition, the mathematical simulation software calculates the area formula of different volume droplet deformation, and the curve was consistent with the trend of force. The comparative experiments and simulations in this article prove that the viscosity and volume of the droplet does not affect the elasticity of the droplet, but the surface tension.

Some further improvements are also required for future work. Firstly, it is necessary to ensure the accuracy of the speed of the pressing plate to reflect the kinetic energy quantity of the droplet converted into the surface energy. Then, it is necessary to repeatedly compress a single droplet to test the probability of its restoration. It is an important property when applied to the elastic droplet in actual production and applications. Eventually, the speed of executing the experiment should be improved to reduce the liquid volatilization.

In summary, droplet elasticity is important in the actual production and application such as targeted delivery and ink-jet printing. The experimental method in this paper provides a better solution for the elasticity measurement of droplets and lays the foundation for future research in this exciting field.

Data accessibility. The data are available from the Dryad Digital Repository: https://doi.org/10.5061/dryad.g1jwstqr0 [38].
Authors' contributions. Y.S.: conceptualization, data curation, formal analysis, funding acquisition, investigation, methodology, software, writing-original draft; M.Z.: formal analysis, funding acquisition, project administration, supervision, writing-review & editing; Y.Z.: formal analysis, methodology, software, supervision, writing-review & editing; L.S.: visualization, writing-review & editing; Y.Z.: writing-review & editing; P.S.: writing-review & editing; CAT.T.: writing-review & editing.
Competing interests. We declare we have no competing interests.
Funding. This work is supported by the National Natural Science Foundation of China under grant nos 52075384 and 51805367.

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
