## [Peer Review File · Royal Society Open Science]

Review History

RSOS-210688.R0 (Original submission)

Review form: Reviewer 1

Is the manuscript scientifically sound in its present form?

Yes

Are the interpretations and conclusions justified by the results?

Yes

Is the language acceptable?

No

Do you have any ethical concerns with this paper?

No

Have you any concerns about statistical analyses in this paper?

No

Recommendation?

Major revision is needed (please make suggestions in comments)

Comments to the Author(s)

This is a manuscript about experiments and modelling of superhydrophobic droplets.

The experimental measurements seem to be valid. The theoretical discussion is reasonable.

However, there are several aspects that must be addressed before the manuscript might be published.

- In general, the presentation of the results lacks clarity as to what scientific or engineering questions are being addressed by these results. Are the authors surprised that "the elasticity of the droplet is provided by surface tension"? This should be expected on very general grounds, I think. The study needs to be clearer about what is novel, and why it is interesting for some branch of science and engineering. Until this is done, a final decision is not possible on whether the paper can be published.

- The references lack volume numbers and page numbers, which is unacceptable.

- The quality of the written English is not high enough for publication, although the meaning is generally clear. This needs to be addressed (I am not sure what is the best mechanism).

- Fig 10 is potentially misleading. The idea seems to be that the damping and viscosity increase together (which should be expected). The scales on the graph axes have been chosen so that the data points for the two quantities are close, but such a choice is only valid if both the left and right versions of the vertical axis start from zero. Indeed one sees that the viscosity grows by a factor approx 60 but the damping only increases by a factor approx 4 -- these quantities are not increasing in proportion to each other, contrary to what the graph seems to indicate. The graph needs to be plotted in a different way that avoids the impression that the two quantities increase in proportion to each other. The connection between damping and viscosity also needs additional theoretical discussion.

- Section 3b claims to have "dynamic" measurements but my impression is that the experiment is actually quasi-static: the 5 second pause after each reduction in height is long enough for the droplet to stop moving. Figure 9 seems to be consistent with this. But if this is true and the experiment is quasi-static, the viscosity should be irrelevant for all the experimental results. On the other hand, the text in Sec 3 seems to imply that viscosity might play a role (although this role is never fully explained).

If the authors want to discuss the observed effects of viscosity in Sec 3, they should first explain how such effects might arise, and what they are supposed to mean. Or if the viscosity is irrelevant because the experiment is quasi-static, this should be stated.

Smaller points

- last sentence of abstract : "the damping ratio increases from ... to ...". It should be stated what quantity is varied as this damping ratio increases.

- section 3(a). I am not sure what is meant by "a puddle shape" for a droplet. Please give a picture or a precise description.

- equation (3.2) has an extra).

- near the end of section 3 : the authors write $40\mu\text{L}$ to $70\mu\text{L}$ but the μ 's should both be greek letters μ .

- a few lines below equation (4.6) there is a broken references (it says "Figure ??"), I think it should be Fig 10.

Review form: Reviewer 2

Is the manuscript scientifically sound in its present form?

Yes

Are the interpretations and conclusions justified by the results?

Yes

Is the language acceptable?

Yes

Do you have any ethical concerns with this paper?

No

Have you any concerns about statistical analyses in this paper?

No

Recommendation?

Major revision is needed (please make suggestions in comments)

Comments to the Author(s)

This manuscript describes experiments and modeling of the squeezing of a drop between two hydrophobic surfaces. The basic phenomenon is well understood and inspiring to the field of microfluidic. I therefore recommend this paper to be published. And it is better if the authors consider the following mentioned remarks and further improve the manuscript before submitting the final version.

1. What is the purity of the glycerol? Please list supplier, grade etc. The presence of any surfactants will affect the results.
2. Page 3: The droplets are not "spherical" as stated above Fig. 1. This is later discussed with respect to the capillary length on page 5. Please be more precise. The capillary length and other properties are defined twice on this page.
3. On page 4: " Three types of droplets were placed on super-hydrophobic surfaces using a precision pipette with volume, V between 5 and 90 μL at 5 μL intervals." The expression is not clear.
4. Incorrect use of parentheses in Eq. (3.2).
5. The author needs to add a space between numbers and units when expressing quantity. On the page 7: " $40\mu\text{L}$ " \square " $40\ \mu\text{L}$ ".
6. Expressing the range of quantity, if the units are of the same magnitude order, only one unit can meet the expression requirements. " $40\ \mu\text{L}$ to $70\ \mu\text{L}$ " \square " 40 to $70\ \mu\text{L}$ "
7. Page 5: Angle representation requires angle system rather than radian system. " π " \square " 180° "
8. Page 6: The detailed expression should be consistent: in the paper, some positions directly use 's' to express time, but some positions use the word "seconds".

9. Some sentences need to be more professional and the description needs to be more accurate. "The small droplet contact area is small, but the surface curvature is big, the internal pressure is big, but the large volume droplet contact area is big, the surface curvature is smaller than the internal pressure."; "but the maximum height difference of different types of droplets is relatively small,".
10. Tense and voice need to be corrected. Page 8: "The absence of an elasticity value on the line segment above does not mean the droplet is crushed, but that the droplet is squeezed away and leaving its original position."; Page 10: "The growth rate also has been increasing with the increasing of damping ratio same as the viscosity."
11. Part of the expression is colloquial and needs to be rigorous. Page 12: " In summary, the understanding of droplet elasticity parameter is of paramount importance in actual production and real life applications such as drug transportation and sensor applications."; "This is an important parameter to consider when applied this parameter in real actual production applications."; "The Eq. (3.2) could explain why the height does not change as the volume increases as the height of the droplet is determined by surface...".
12. The reference of the figure 11b is miss in the text." the fitting curve of the elastic force is shown in Figure ??b".
13. Comma is missing after reference 25 on page 2.
14. Repeated expression meaning within one statement. Page 2: "Here, droplets of different volumes and concentrations were placed between two parallel plates of super-hydrophobic substrates."

Decision letter (RSOS-210688.R0)

Dear Dr Zhao,

The Editors assigned to your paper RSOS-210688 "Elasticity and damping ratio measurement of droplets on super-hydrophobic surfaces" have made a decision based on their reading of the paper and any comments received from reviewers.

Regrettably, in view of the reports received, the manuscript has been rejected in its current form. However, a new manuscript may be submitted which takes into consideration these comments.

We invite you to respond to the comments supplied below and prepare a resubmission of your manuscript. Below the referees' and Editors' comments (where applicable) we provide additional requirements. We provide guidance below to help you prepare your revision.

Please note that resubmitting your manuscript does not guarantee eventual acceptance, and we do not generally allow multiple rounds of revision and resubmission, so we urge you to make every effort to fully address all of the comments at this stage. If deemed necessary by the Editors, your manuscript will be sent back to one or more of the original reviewers for assessment. If the original reviewers are not available, we may invite new reviewers.

Please resubmit your revised manuscript and required files (see below) no later than 27-Mar-2022. Note: the ScholarOne system will 'lock' if resubmission is attempted on or after this deadline. If you do not think you will be able to meet this deadline, please contact the editorial office immediately.

Please note article processing charges apply to papers accepted for publication in Royal Society Open Science (<https://royalsocietypublishing.org/rsos/charges>). Charges will also apply to papers transferred to the journal from other Royal Society Publishing journals, as well as papers submitted as part of our collaboration with the Royal Society of Chemistry (<https://royalsocietypublishing.org/rsos/chemistry>). Fee waivers are available but must be requested when you submit your manuscript (<https://royalsocietypublishing.org/rsos/waivers>).

Thank you for submitting your manuscript to Royal Society Open Science and we look forward to receiving your resubmission. If you have any questions at all, please do not hesitate to get in touch.

on behalf of Dr David Wales (Associate Editor) and R. Kerry Rowe (Subject Editor)
openscience@royalsociety.org

Associate Editor Comments to Author (Dr David Wales):

Comments to the Author:

Both referees have concluded that major revisions are required before this manuscript might be suitable for publication. It is not clear whether a revised version will actually be acceptable. Hence the manuscript is rejected at this stage, but resubmission would be permitted.

Reviewer comments to Author:

Reviewer: 1

Comments to the Author(s)

This is a manuscript about experiments and modelling of superhydrophobic droplets.

The experimental measurements seem to be valid. The theoretical discussion is reasonable.

However, there are several aspects that must be addressed before the manuscript might be published.

- In general, the presentation of the results lacks clarity as to what scientific or engineering questions are being addressed by these results. Are the authors surprised that "the elasticity of the droplet is provided by surface tension"? This should be expected on very general grounds, I think. The study needs to be clearer about what is novel, and why it is interesting for some branch of science and engineering. Until this is done, a final decision is not possible on whether the paper can be published.

- The references lack volume numbers and page numbers, which is unacceptable.

- The quality of the written English is not high enough for publication, although the meaning is generally clear. This needs to be addressed (I am not sure what is the best mechanism).

- Fig 10 is potentially misleading. The idea seems to be that the damping and viscosity increase together (which should be expected). The scales on the graph axes have been chosen so that the data points for the two quantities are close, but such a choice is only valid if both the left and

right versions of the vertical axis start from zero. Indeed one sees that the viscosity grows by a factor approx 60 but the damping only increases by a factor approx 4 -- these quantities are not increasing in proportion to each other, contrary to what the graph seems to indicate. The graph needs to be plotted in a different way that avoids the impression that the two quantities increase in proportion to each other. The connection between damping and viscosity also needs additional theoretical discussion.

- Section 3b claims to have "dynamic" measurements but my impression is that the experiment is actually quasi-static: the 5 second pause after each reduction in height is long enough for the droplet to stop moving. Figure 9 seems to be consistent with this. But if this is true and the experiment is quasi-static, the viscosity should be irrelevant for all the experimental results. On the other hand, the text in Sec 3 seems to imply that viscosity might play a role (although this role is never fully explained).

If the authors want to discuss the observed effects of viscosity in Sec 3, they should first explain how such effects might arise, and what they are supposed to mean. Or if the viscosity is irrelevant because the experiment is quasi-static, this should be stated.

Smaller points

- last sentence of abstract : "the damping ratio increases from ... to ...". It should be stated what quantity is varied as this damping ratio increases.

- section 3(a). I am not sure what is meant by "a puddle shape" for a droplet. Please give a picture or a precise description.

- equation (3.2) has an extra).

- near the end of section 3 : the authors write $40\mu\text{L}$ to $70\mu\text{L}$ but the μ 's should both be greek letters μ .

- a few lines below equation (4.6) there is a broken references (it says "Figure ??"), I think it should be Fig 10.

Reviewer: 2

Comments to the Author(s)

This manuscript describes experiments and modeling of the squeezing of a drop between two hydrophobic surfaces. The basic phenomenon is well understood and inspiring to the field of microfluidic. I therefore recommend this paper to be published. And it is better if the authors consider the following mentioned remarks and further improve the manuscript before submitting the final version.

1. What is the purity of the glycerol? Please list supplier, grade etc. The presence of any surfactants will affect the results.
2. Page 3: The droplets are not "spherical" as stated above Fig. 1. This is later discussed with respect to the capillary length on page 5. Please be more precise. The capillary length and other properties are defined twice on this page.
3. On page 4: " Three types of droplets were placed on super-hydrophobic surfaces using a precision pipette with volume, V between 5 and 90 μL at 5 μL intervals." The expression is not clear.
4. Incorrect use of parentheses in Eq. (3.2).
5. The author needs to add a space between numbers and units when expressing quantity. On the page 7: " $40\mu\text{L}$ " \square " $40\ \mu\text{L}$ ".

6. Expressing the range of quantity, if the units are of the same magnitude order, only one unit can meet the expression requirements. "40 uL to 70 uL" □ "40 to 70 uL"
7. Page 5: Angle representation requires angle system rather than radian system. " π " □ "180°"
8. Page 6: The detailed expression should be consistent: in the paper, some positions directly use 's' to express time, but some positions use the word "seconds".
9. Some sentences need to be more professional and the description needs to be more accurate. "The small droplet contact area is small, but the surface curvature is big, the internal pressure is big, but the large volume droplet contact area is big, the surface curvature is smaller than the internal pressure."; "but the maximum height difference of different types of droplets is relatively small,".
10. Tense and voice need to be corrected. Page 8: "The absence of an elasticity value on the line segment above does not mean the droplet is crushed, but that the droplet is squeezed away and leaving its original position."; Page 10: "The growth rate also has been increasing with the increasing of damping ratio same as the viscosity."
11. Part of the expression is colloquial and needs to be rigorous. Page 12: " In summary, the understanding of droplet elasticity parameter is of paramount importance in actual production and real life applications such as drug transportation and sensor applications."; "This is an important parameter to consider when applied this parameter in real actual production applications."; "The Eq. (3.2) could explain why the height does not change as the volume increases as the height of the droplet is determined by surface...".
12. The reference of the figure 11b is miss in the text." the fitting curve of the elastic force is shown in Figure ??b".
13. Comma is missing after reference 25 on page 2.
14. Repeated expression meaning within one statement. Page 2: "Here, droplets of different volumes and concentrations were placed between two parallel plates of super-hydrophobic substrates."

===PREPARING YOUR MANUSCRIPT===

If you have been asked to revise the written English in your submission as a condition of publication, you must do so, and you are expected to provide evidence that you have received language editing support. The journal would prefer that you use a professional language editing

service and provide a certificate of editing, but a signed letter from a colleague who is a native speaker of English is acceptable. Note the journal has arranged a number of discounts for authors using professional language editing services (<https://royalsociety.org/journals/authors/benefits/language-editing/>).

===PREPARING YOUR REVISION IN SCHOLARONE===

-- If you have uploaded ESM files, please ensure you follow the guidance at <https://royalsociety.org/journals/authors/author-guidelines/#supplementary-material> to include a suitable title and informative caption. An example of appropriate titling and captioning

may be found at [https://figshare.com/articles/Table_S2_from_Is_there_a_trade-off_between_peak_performance_and_performance_breadth_across_temperatures_for_aerobic_sc
ope_in_teleost_fishes_/3843624](https://figshare.com/articles/Table_S2_from_Is_there_a_trade-off_between_peak_performance_and_performance_breadth_across_temperatures_for_aerobic_scope_in_teleost_fishes_/3843624).

Author's Response to Decision Letter for (RSOS-210688.R0)

See Appendix A.

RSOS-211632.R0

Review form: Reviewer 1

Is the manuscript scientifically sound in its present form?

Yes

Are the interpretations and conclusions justified by the results?

Yes

Is the language acceptable?

Yes

Do you have any ethical concerns with this paper?

No

Have you any concerns about statistical analyses in this paper?

No

Recommendation?

Accept with minor revision (please list in comments)

Comments to the Author(s)

The manuscript has been improved in this resubmission. In particular, the first part of the resubmission letter now explains the motivation for this work (which was not explained properly in the initial manuscript). The standard of written English is much improved, there are still a few minor errors but I would expect that these can be fixed at the copy editing stage.

The paper can be accepted, I have only one condition:

The supporting data that I downloaded consists of 3 .rar files. In addition to this data, there must be an additional human-readable file (eg in pdf or raw text format) with instructions as to how to open the .rar files, and gives a summary of what they contain. I believe that I raised this point in my original review but it does not seem to have been answered.

Decision letter (RSOS-211632.R0)

Dear Dr zhao

On behalf of the Editors, we are pleased to inform you that your Manuscript RSOS-211632 "Elasticity and damping ratio measurement of droplets on super-hydrophobic surfaces" has been accepted for publication in Royal Society Open Science subject to minor revision in accordance with the referees' reports. Please find the referees' comments along with any feedback from the Editors below my signature.

Please submit your revised manuscript and required files (see below) no later than 7 days from today's (ie 01-Dec-2021) date. Note: the ScholarOne system will 'lock' if submission of the revision is attempted 7 or more days after the deadline. If you do not think you will be able to meet this deadline please contact the editorial office immediately.

on behalf of Dr David Wales (Associate Editor) and R. Kerry Rowe (Subject Editor)
openscience@royalsociety.org

Associate Editor Comments to Author (Dr David Wales):

Associate Editor

Comments to the Author:

The revisions have largely addressed the concerns of the referee. A revised manuscript that satisfies the remaining condition of this referee can be accepted.

Reviewer comments to Author:

Reviewer: 1

Comments to the Author(s)

The manuscript has been improved in this resubmission. In particular, the first part of the resubmission letter now explains the motivation for this work (which was not explained properly

in the initial manuscript). The standard of written English is much improved, there are still a few minor errors but I would expect that these can be fixed at the copy editing stage.

The paper can be accepted, I have only one condition:

The supporting data that I downloaded consists of 3 .rar files. In addition to this data, there must be an additional human-readable file (eg in pdf or raw text format) with instructions as to how to open the .rar files, and gives a summary of what they contain. I believe that I raised this point in my original review but it does not seem to have been answered.

===PREPARING YOUR MANUSCRIPT===

one version should clearly identify all the changes that have been made (for instance, in coloured highlight, in bold text, or tracked changes);

===PREPARING YOUR REVISION IN SCHOLARONE===

-- If you are requesting an article processing charge waiver, you must select the relevant waiver option (if requesting a discretionary waiver, the form should have been uploaded, see 'File upload' above).

-- If you have uploaded any electronic supplementary (ESM) files, please ensure you follow the guidance at <https://royalsociety.org/journals/authors/author-guidelines/#supplementary-material> to include a suitable title and informative caption. An example of appropriate titling and captioning may be found at https://figshare.com/articles/Table_S2_from_Is_there_a_trade-off_between_peak_performance_and_performance_breadth_across_temperatures_for_aerobic_scope_in_teleost_fishes_/3843624.

Author's Response to Decision Letter for (RSOS-211632.R0)

See Appendix B.

Decision letter (RSOS-211632.R1)

Dear Dr zhao,

I am pleased to inform you that your manuscript entitled "Elasticity and damping ratio measurement of droplets on super-hydrophobic surfaces" is now accepted for publication in Royal Society Open Science.

on behalf of Dr David Wales (Associate Editor) and R. Kerry Rowe (Subject Editor)
openscience@royalsociety.org

Appendix A

Dear editor of *Royal society open science*,

We *resubmit* a revised manuscript titled “Elasticity and damping ratio measurement of droplets on super-hydrophobic surfaces” to the journal of *Royal society open science*.

Thank you very much for the reviewer’s comments concerning our manuscript entitled “Elasticity and damping ratio measurement of droplets on super-hydrophobic surfaces”. Those comments are all valuable and helpful for improving the quality of our work. We have made correction which we hope meet with approval.

Thanks a lot for considering our manuscript to be published on the journal of *Royal society open science*.

Yours sincerely!

Dr. Meirong Zhao

State Key Laboratory of Precision Measuring Technology and Instruments

Tianjin University

Tianjin 300072, China.

Reviewer 1

1. In general, the presentation of the results lacks clarity as to what scientific or engineering questions are being addressed by these results. Are the authors surprised that "the elasticity of the droplet is provided by surface tension"? This should be expected on very general grounds, I think. The study needs to be clearer about what is novel, and why it is interesting for some branch of science and engineering. Until this is done, a final decision is not possible on whether the paper can be published.

Re: The droplet elasticity is of fundamental importance in many branches of engineering. For example, these are the activities in the manufacturing of flat displays by ink jet printing and plastic electronics. Currently, there exist two main methods for measuring droplet elasticity: capillary micromechanics (Fig.1) and droplet impacting (Fig.2).

Fig.1 Capillary micromechanics

Fig.2 Droplet impacting

But the both methods have shortcomings. The methods of capillary micromechanics can only be used to test small droplets. The second method will lose accuracy when it works under large Weber number. For example, droplet impacts on rigid surfaces define six possible scenarios for such impingements, including deposition, prompt splash, corona splash, receding break-up, partial rebound and complete rebound. The inertial energy is converted to interfacial energy causing the drop to spread and deform. As the impact velocity increases (higher Weber number), more kinetic energy is spent for droplet deformation and less energy remains for bouncing. Moreover, the droplet can be broken up and lost some liquid due to high deformations (Fig.3).

Fig.3 Prompt splash

Our innovative experimental method is relatively simple, without the limitation of volume and Weber number. Meanwhile, it also proves by experiments that the elasticity of droplets is provided by surface tension and independent of volume and viscosity. Different viscosity will only interfere with its damping ratio and the time it takes to reach stability. Those results (the physics of the droplet) can be used for inkjet printing and droplet-based microfluidic.

These statements had been added in abstract (page 1), discussion (page 9 and 10) and conclusion (page 11) of the newly revised manuscript.

2. The references lack volume numbers and page numbers, which is unacceptable.

Re: Volume numbers, page numbers and doi codes had been added to the references (page 12) in the revised manuscript.

3. The quality of the written English is not high enough for publication, although the meaning is generally clear. This needs to be addressed (I am not sure what is the best mechanism).

Re: Errors and unreasonable expressions had been changed.

4. Fig 10 is potentially misleading. The idea seems to be that the damping and viscosity increase together (which should be expected). The scales on the graph axes have been chosen so that the data points for the two quantities are close, but such a choice is only valid if both the left and right versions of the vertical axis start from zero. Indeed, one sees that the viscosity grows by a factor approx 60 but the damping only increases by a factor approx 4 -- these quantities are not increasing in proportion to each other, contrary to what the graph seems to indicate. The graph needs to be plotted in a different way that avoids the impression that the two quantities increase in proportion to each other. The connection between damping and viscosity also needs additional theoretical discussion.

Re: Fig 10 had been corrected (Fig.4). Both the vertical axes started from zero.

Fig.4 The standard mass-spring-damper (MSD) model and the relationship between damping ratio and droplet viscosity.

When the platen contacted with the droplet and stopped extrusion, its trampoline response can be considered as a standard mass-spring-damper (MSD) system. The data (speed of inner liquid, Fig.5) in this process was obtained through simulation experiments, but it was not described in this paper. The simulation experiments had been added to paragraph 2 in the section 4(b) page 10 of the newly revised manuscript.

Fig.5 Trampoline response

5. Section 3b claims to have "dynamic" measurements but my impression is that the experiment is actually quasi-static: the 5 second pause after each reduction in height is long enough for the droplet to stop moving. Figure 9 seems to be consistent with this. But if this is true and the experiment is quasi-static, the viscosity should be irrelevant for all the experimental results. On the other hand, the text in Sec 3 seems to imply that viscosity might play a role (although this role is never fully explained).

If the authors want to discuss the observed effects of viscosity in Sec 3, they should first explain how such effects might arise, and what they are supposed to mean. Or if the viscosity is irrelevant because the experiment is quasi-static, this should be stated.

Re: For a single measurement, it is indeed quasi-static. It only spends 0.2 s to reach the stable equilibrium state in the simulation experiment (Figure 9 and Fig.5). In my paper, the dynamics that we want to express refers to the motion of the upper pressing plate in the whole droplet extrusion process (step shape, Fig.6), rather than just once. It aims to traverse each extrusion state of the droplet and change the droplet area. Match the mathematical model with the measured data.

Viscosity is mainly used as the variable of simulation experiment, and the simulation data proves that viscosity affects the damping ratio and the time to reach stability rather than elasticity.

The unreasonable expression "dynamic" had been changed in section 3(b) page 6. And the simulation experiment of viscosity had been added to paragraph 2 in the section 4(b) page 10 of the newly revised manuscript.

Fig.6 The motion of the upper pressing plate

6. Smaller points

- last sentence of abstract: "the damping ratio increases from ... to ...". It should be stated what quantity is varied as this damping ratio increases.

Re: The quantity is mass fraction and the correct expression had been added in the abstract (page 1).

- section 3(a). I am not sure what is meant by "a puddle shape" for a droplet. Please give a picture or a precise description.

Re: The picture is Figure 3 and the cite had been added after "a puddle shape" (paragraph 1 in page 5). The droplet in fig.7 can be considered as quasi-spherical. Fig.8 describes a droplet which is a puddle shape. The definition of "a puddle shape" could be found in the literatures:

(1) *Surface tension of liquid marbles.*

(2) *Properties of liquid marbles.*

(3) *Shape, vibrations, and effective surface tension of water marbles.*

Fig.7 3 μ L droplet

Fig.8 45 μ L droplet

- equation (3.2) has an extra).

Re: Extra parentheses had been deleted (page 5).

- near the end of section 3: the authors write 40 μ L to 70 μ L but the μ 's should both be Greek letters μ .

Re: The error had been corrected (paragraph 5 in page 6).

- a few lines below equation (4.6) there is a broken references (it says "Figure ??"), I think it should be Fig 10.

Re: The broken reference had been corrected (paragraph 4 in page 10).

Reviewer 2

1. What is the purity of the glycerol? Please list supplier, grade etc. The presence of any surfactants will affect the results.

Re: The purity of the glycerol is AR. And it was bought from Shanghai Hushi Laboratorial Equipment Co., Ltd. The missing information had been supplemented in the revised manuscript (paragraph 2 of section 2).

2. Page 3: The droplets are not "spherical" as stated above Fig. 1. This is later discussed with respect to the capillary length on page 5. Please be more precise. The capillary length and other properties are defined twice on this page.

Re: The capillary length defined twice had been changed to once (paragraph 2 in page 5).

3. On page 4: "Three types of droplets were placed on super-hydrophobic surfaces using a precision pipette with volume, V between 5 and 90 μ L at 5 μ L intervals." The expression is not clear.

Re: The expression had been changed to: "Three types of droplets were placed on super-hydrophobic surfaces using a precision pipette. And the volume from 5 to 90 μ L at 5 μ L intervals." (paragraph 2 in page 4).

4. Incorrect use of parentheses in Eq. (3.2).

Re: Extra parentheses had been deleted (page 5).

5. The author needs to add a space between numbers and units when expressing quantity. On the page 7: "40uL" → "40 uL".

Re: The space between numbers and units had been added in the revised manuscript. The quantity expressing in all paper had been changed.

6. Expressing the range of quantity, if the units are of the same magnitude order, only one unit can meet the expression requirements. "40 uL to 70 uL" → "40 to 70 uL"

Re: This type error in all paper had been detected and corrected.

7. Page 5: Angle representation requires angle system rather than radian system. " π " → "180°"

Re: This error had been detected and corrected in the revised manuscript (paragraph 2 in page 5).

8. Page 6: The detailed expression should be consistent: in the paper, some positions directly use 's' to express time, but some positions use the word "seconds".

Re: This type error had been detected and corrected in the revised manuscript. All units of time are unified into 's'.

9. Some sentences need to be more professional and the description needs to be more accurate. "The small droplet contact area is small, but the surface curvature is big, the internal pressure is big, but the large volume droplet contact area is big, the surface curvature is smaller than the internal pressure."; "but the maximum height difference of different types of droplets is relatively small,".

Re: Unprofessional sentences and expressions have been changed. "The contact area of small droplets is different from large droplets when they are compressed at a certain distance. For example, the 40 uL droplet contact area is smaller than 70 uL droplet." (paragraph 7 in page 6), "but the maximum height difference of different types of droplets can be ignored" (paragraph 1 in page 5)

10. Tense and voice need to be corrected. Page 8: "The absence of an elasticity value on the line segment above does not mean the droplet is crushed, but that the droplet is squeezed away and leaving its original position."; Page 10: "The growth rate also has been increasing with the increasing of damping ratio same as the viscosity."

Re: Incorrect tenses and voices had been updated. "When the force was sufficient to overcome the friction between the plates and droplet, it would leave the original position and the indication decreases sharply, which marks the end of the extrusion process." (paragraph 4 in page 8), "When the mass fraction of liquid increased from 0 to 80 wt% (Figure 10), the damping ratio increased from 0.068 to 0.261." (paragraph 4 in page 10).

11. Part of the expression is colloquial and needs to be rigorous. Page 12: "In summary, the understanding of droplet elasticity parameter is of paramount importance in actual production and real life applications such as drug transportation and sensor applications."; "This is an important parameter to consider when applied this parameter in real actual production applications."; "The Eq. (3.2) could explain why the height does not change as the volume increases as the height of the droplet is determined by surface...".

Re: The unreasonable expression had been changed to: “In summary, the droplet elasticity is important in the actual production and application such as drug transportation and sensor applications.” (paragraph 3 in page 11), “It is an important property when applied the elastic droplet in actual production and applications.” (paragraph 2 in page 11), “The Eq. (3.2) could explain why the height does not change with the volume increases, since the height of the droplet is determined by surface tension, density, and gravitational acceleration. But has no relationship with volume.” (paragraph 1 in page 6).

12. The reference of the figure 11b is miss in the text.” the fitting curve of the elastic force is shown in Figure ??b”.

Re: This error had been detected and corrected in the revised manuscript (paragraph 4 in page 10).

13. Comma is missing after reference 25 on page 2.

Re: The comma had been added in the revised manuscript (paragraph 3 in page 2).

14. Repeated expression meaning within one statement. Page 2: “Here, droplets of different volumes and concentrations were placed between two parallel plates of super-hydrophobic substrates.”

Re: The wrong expression had been changed to: “Here, droplets of different volumes and viscosities were placed between two parallel plates of super-hydrophobic substrates.” (paragraph 3 in page 2).

Appendix B

Dear editor of *Royal society open science*,

We *resubmit* a revised manuscript titled “Elasticity and damping ratio measurement of droplets on super-hydrophobic surfaces” to the journal of *Royal society open science*.

Thank you very much for the reviewer’s comments concerning our manuscript entitled “Elasticity and damping ratio measurement of droplets on super-hydrophobic surfaces”. Those comments are all valuable and helpful for improving the quality of our work. We have made corrections which we hope to meet with approval.

Thanks a lot for considering our manuscript to be published in the journal of *Royal society open science*.

Yours sincerely!

Dr. Meirong Zhao

State Key Laboratory of Precision Measuring Technology and Instruments

Tianjin University

Tianjin 300072, China.

Reviewer 1

1. The manuscript has been improved in this resubmission. In particular, the first part of the resubmission letter now explains the motivation for this work (which was not explained properly in the initial manuscript). The standard of written English is much improved, there are still a few minor errors but I would expect that these can be fixed at the copy-editing stage.

The paper can be accepted, I have only one condition:

The supporting data that I downloaded consists of 3 .rar files. In addition to this data, there must be an additional human-readable file (eg in pdf or raw text format) with instructions as to how to open the .rar files, and gives a summary of what they contain. I believe that I raised this point in my original review but it does not seem to have been answered.

Re: Some minor errors in the paper have been corrected and marked in the Highlighted version. A PDF file, *Instructions of ESM*, has been added to the ESM folder to explain the experimental data.

Instructions of ESM

The ESM file contains three RAR files: *01 Side_and_top_view_of_droplets*, *02 Elasticity_measurement* and *03 Simulation_data*.

1. The first file, *01 Side_and_top_views_of_droplets*, contains side views and top views of different volume droplets. These pictures are processed through an image algorithm to generate the curves in Fig. 4 of the paper.
2. The second file, *02 Elasticity_measurement*, has two subfiles, which are *pic* and *compressed data*. The pictures in the *pic* subfile show the compression of droplets during the experiment. The Excel file, *compressed data*, contains 3 sheets:
 - 1) *01 Balance data*: the data from this sheet are the values of the electronic balance showing elasticity measurements of different volume droplets. These data can be described in Fig. 5.
 - 2) *02 Force value*: the data from this sheet record the elastic force values of droplets (the gravity of the droplet itself not included). This sheet was summarized in Fig 8.
 - 3) *03 Ball crown*: the data from this sheet are the droplet geometric parameters calculated by the compression model, which can calculate the change of droplet surface area during compression (Fig. 7 and Fig. 11).
3. The last file, *03 Simulation_data*, has five folders and two Excel files. The five folders record the internal pressure of different mass fractions droplets. The two Excel files separately record the damping ratio and the oscillation time of different mass fractions droplets. These data are obtained through simulation (Fig. 9 and Fig. 10).